# Role of Context in Unsupervised Sentence Representation Learning: the Case of Dialog Act Tagging

**Rastislav Hronsky**
Jheronimus Academy of Data Science
Sint Janssingel 92, 5211 DA 's-Hertogenbosch
Netherlands
r.hronsky@tue.nl

**Emmanuel Keuleers**
Tilburg University
Warandelaan 2, 5037 AB Tilburg
Netherlands
E.A.Keuleers@tilburguniversity.edu

## Abstract

Unsupervised learning of word representations involves capturing the contextual information surrounding word occurrences, which can be grounded in the observation that word form is largely disconnected from word meaning. While there are fewer reasons to believe that the same holds for sentences, learning through context has been carried over to learning representations of word sequences. However, this work pays minimal to no attention to the role of context in inferring sentence representations. In this article, we present a dialog act tag probing task designed to explicitly compare content-, and context-oriented sentence representations inferred on utterances of telephone conversations (SwDA). Our results suggest that there is no clear benefit of context-based sentence representations over content-based sentence representations. However, there is a very clear benefit of increasing the dimensionality of the sentence vectors in nearly all approaches.

## 1 Introduction

Unsupervised methods for constructing word representations vary widely in their implementation (Landauer and Dumais, 1997; Mikolov et al., 2013a; Pennington et al., 2014), but share the core property that they try to capture the contextual information surrounding word occurrences. This can be grounded in the observation that word form is largely disconnected from word meaning (De Saussure, 1916) and that, therefore, trying to extract meaning from form directly would be pointless.

However, there are fewer reasons to believe that representing the meaning of a fully composed sentence requires the description of its *context* in the way that a word does. Still, much effort has been devoted to developing and evaluating context-oriented methods for sentence representation (Kiros et al., 2015; Logeswaran and Lee, 2018; Cer et al., 2018). Interestingly, the theoretical justification for the approach –roughly paraphrased– is

that it works well for words (Kiros et al., 2015).

In the present article, we examine whether a contextual signal is of any significant benefit to general-purpose sentence representation.

To accommodate this goal, we infer two sets of *utterance* representations from telephone conversation transcripts, one by a model trained to encode the transcripts' *content*, and one by a model trained to encode the transcripts *context* in a Skip-Thought-like fashion. We evaluate the learned representations with a *linear probe* classifying the dialog act tags. Below, we discuss these choices more extensively.

The reason for the *linear probing* technique is to avoid any further representation learning in the downstream task training, such that the resulting accuracy numbers solely reflect the unsupervised representations. This technique is commonly used to assess to what extent sentence representations exhibit certain linguistic properties, such as part-of-speech or syntax (Conneau et al., 2018; Chrupała and Alishahi, 2019; Perone et al., 2018; Belinkov, 2022a).

We chose the *Switchboard Dialog Act Corpus* (SwDA) (Jurafsky, 1997) corpus for the following properties: (1) its size allows for inferring representations with both traditional, statistical learning methods and modern neural network based methods; (2) at the same time, the corpus is fairly narrow-domain, enabling small models to converge to meaningful representations [1]; (3) it contains utterance labels that we consider suitable for a probing task with respect to contextuality: while a dialog act *tag* unquestionably depends on the utterance contents, the surrounding utterances likely provide a good signal about its label too. See example utterances from the corpus in Table 1.

---

[1] As opposed to some other well-known sentence-level tasks, such as NLI (Bowman et al., 2015), SICK (Marelli et al., 2014), SST-2 (Socher et al., 2013), that require "universal" sentence representations.

| SD^E | *But I, my grandparents were looking into it before.* |
| --- | --- |
| SD | *So I know what they've said.* |
| B | *Uh-huh.* |
| ^H | *Well, I'm trying to think.* |

Table 1: Example dialog act tags and utterances from the SwDA corpus. (SD$^\wedge$E = *Statement expanding y/n answer*, SD = *Statement-non-opinion*, B = *Backchannel*, $^\wedge$H = *Hold before answer/agreement*).

Using context to enhance dialog act predictions is not a novel idea (Raheja and Tetreault, 2019; Li and Wu, 2019; Kalchbrenner and Blunsom, 2013). To ascertain the usefulness of context in this task, we checked for the reduction in entropy of a tag in a "unigram" setting vs. conditioned on a previous tag, a "bigram" setting. The results were $H(currenttag) = 2.1bit$ and $H(currenttag|previoustag) = 1.86bit$, meaning that such a context-enhanced baseline would, at least, increase the chance-level accuracy.

For the *content-oriented* representation variants, the representations of utterances were inferred via classic topic modeling methods, word vector averaging, or an LSTM-based auto-encoder (see Section 3 for details). The resulting sentence vectors were then used as input for a Skip-Thought-like training regime, where the goal was to learn their *context-oriented* version by projecting them such that they are predictive of surrounding utterances. Additionally, we trained a version where the utterance encoding and projection are trained jointly, with an LSTM-based encoder of identical proportions with the one used for auto-encoding.

## 2 Related Work

In the literature review, we summarize some of the most important work in sentence representation and relate it to the issues outlined in previous paragraphs.

Before the deep learning era of NLP, text-processing methods relied mainly on lexical features of sentences and were invariant to word order. Using word frequency transformations, e.g., *tf-idf* (Ramos et al., 2003), has proven effective to base the representations on more semantically laden words. Techniques such as *singular value decomposition* (SVD) (Landauer and Dumais, 1997; Landauer et al., 1998), or topic modeling (Blei et al., 2003; Blei, 2012; Hofmann, 2001) were used

to transform the sparse word-count matrices into dense vectors.

With the subsequent success of simple neural networks such as *word2vec* (Mikolov et al., 2013a,b; Goldberg and Levy, 2014) or *GloVe* (Pennington et al., 2014) at representing words and phrases, a seemingly naive method of averaging these word vectors was shown to produce useful sentence/paragraph/document representations (Arora et al., 2017).

With further success of neural networks in text processing, the research on the topic grew markedly. Among the unsupervised methods, recursive auto-encoders (RAEs) (Socher et al., 2011a,b; Pollack, 1990) were trained to reconstruct sentences via recursively encoding nodes in a dependency parse tree and performed well at a paraphrasing task (Dolan et al., 2004). *Tree-LSTM*s (Tai et al., 2015) later succeeded the RAEs in the same recursive set up but using an LSTM encoder (Hochreiter and Schmidhuber, 1997), and evaluated on predicting semantic relatedness between two sentences (Marelli et al., 2014) and classifying sentiment (Socher et al., 2013). The current state-of-the-art unsupervised, general-purpose sentence representation systems are based on pre-trained Transformer language models (Vaswani et al., 2017) and fine-tuned via *contrastive learning* (Gutmann and Hyvärinen, 2012; Chen et al., 2020), i.e., pulling representations of alternative sentence formulations closer together and those of unrelated, random sentence pairs apart (Gao et al., 2021; Chuang et al., 2022).

The success of skip-gram-based word representations inspired researchers to transfer the idea to sentence representation. *Skip-Thought* (Kiros et al., 2015) is a large, LSTM-based (Hochreiter and Schmidhuber, 1997), sentence encoder that was trained to produce representations predictive of surrounding sentences. *Quick-Thought* (Logeswaran and Lee, 2018) is a similar system that differs in that it casts the problem as *contrastive learning* (Gutmann and Hyvärinen, 2012). The *Universal Sentence Encoder* (USE) is a Transformer-based (Vaswani et al., 2017) system introduced by Cer et al. (2018) that uses the Skip-Thought as one of the tasks in a multi-task learning set-up (including some supervised tasks). The task of deciding whether two sentences are consecutive or not (NSP task) was also used in training of *BERT* (Devlin et al., 2019). The latest dialog processing sys-

tems, built on top of Transformer language models (Vaswani et al., 2017), typically learn to predict contextual utterances within the contrastive learning framework and achieve superior performance on dialog tasks compared to universal representations (Wu et al., 2020; Zhou et al., 2022; Zeng et al., 2023; Oh et al., 2023).

While sentence contextuality is used in all of the above-mentioned studies, there is little to no focus on assessing the necessity for using context to derive sentence representations. An ablation of the NSP task is presented by Devlin et al. (2019), suggesting a slight performance benefit over a version where the task is omitted. However, Devlin et al. (2019) only evaluate the fine-tuning pipeline, while the raw BERT representations were shown to perform poorly overall (Reimers and Gurevych, 2019). Further, results by Liu et al. (2019) are contradictory and do not credit a performance improvement to the NSP task. Lastly, Zhou et al. (2022) additionally compare the representations resulting from dropout augmentations (Gao et al., 2021) to those of consecutive utterrances and found a significant benefit in favor of the latter augmentations. Their results are perplexing, however, since the dropout version they implemented performed even worse than that of *SimCSE* (Gao et al., 2021), which also uses dropout augmentation, and should in fact perform worse, not being tailored for dialog.

Using the natural language inference (NLI) task has been effective in several general sentence representation learning systems (Conneau et al., 2017; Reimers and Gurevych, 2019; Zhou et al., 2022). However, along with machine translation models, these systems fall outside the category of unsupervised sentence representations.

## 3 Method

We trained several variants of content-based representations, their context-based counterparts, and an end-to-end, Skip-Thought like system, trained to encode and predict sentences.

We evaluated the following variants of *content-based* sentence representation [2]:

1. Latent Semantic Analysis (LSA) $^\star$

2. Latent Dirichlet Allocation (LDA)$^\star$ (Blei et al., 2003)

3. Mean of GloVe word vectors $^\star$ (Pennington et al., 2014) [3]

4. LSTM-based auto-encoder (Hochreiter and Schmidhuber, 1997)

5. Mean of arbitrarily assigned word vectors

We also varied the final dimension of the sentence vector, namely $D \in \{6, 12, 25\}$, either by setting the number_of_topics parameter, setting the auto-encoder bottleneck size, or truncating GloVe vectors. The auto-encoder was based on a 2-layer, bidirectional Long Short-Term Memory (LSTM) neural network with hidden size of $D_{hidden} = 64$, dropout of $p = 0.1$. The variant 5 is one where we randomly assigned a feature vector from an 8-dimensional raster to each vocabulary entry (a procedure by which we want to emulate the usage of 1-hot vector encoding, but with dense, real-valued vectors), and created the final sentence representation by averaging corresponding word vectors.

To create the *context-based* representations, we devised a learning regimen for 2 linear layers for each variant. One layer was used to project the sentence content vector to the context space (encode), and the other layer was used to predict the content vectors of surrounding sentences from the context vector (decode).

Finally, we trained a Skip-Thought-like model, where each sentence was represented by a sequence of word vectors (obtained via the same procedure as in variant 5), optionally truncated and padded to 20 tokens, processed via an LSTM (identical with the auto-encoder in variant 4), and projected such that, conditioned on the resulting vector, a decoder-LSTM reconstructs the surrounding sentences well via *teacher forcing*.

In all of the context-based variants, we varied the context size between 1 and 2 sentences, on each side.

### 3.1 Dataset and Evaluation

To infer and evaluate the representations, we used the *Switchboard Dialog Act Corpus* (SwDA) (Jurafsky, 1997), with exception of the GloVe vectors, where we used the pre-trained *glove-twitter-25* vectors. The SwDA contains transcripts of casual telephone conversations in which the two parties were instructed to discuss a certain topic, e.g. "care of the elderly". Every utterance is assigned a label

---

[2]Entries marked with $\star$ use the GenSim implementation.

[3]We used the pre-trained *glove-twitter-25* version.

|                | Train    | Dev     | Test    |
|----------------|----------|---------|---------|
| Utterances     | 212,145  | 26,518  | 26,519  |
| Tokens         | 1,833M   | 227K    | 227K    |
| Tokens per utt.| 8.64     | 8.56    | 8.32    |
| SD             | 35.47%   | 33.52%  | 35.00%  |
| %/Other        | 27.45%   | 25.77%  | 25.60%  |
| B              | 18.52%   | 21.54%  | 23.02%  |
| SV             | 13.76%   | 14.60%  | 13.73%  |
| AA             | 4.81%    | 4.57%   | 2.65%   |

Table 2: Summary of the SwDA dataset. In the upper section of the table, we summarize the corpus statistics. In the lower section we list the percentages of dialog act labels per corpus split (*SD*="Statement-non-opinion", *B*="Backchannel", *SV*="Statement-opinion", *AA*="Agree/Accept").

describing the "role" of the act, in a similar way that part-of-speech tags describe roles of words in sentences [4].

To pre-process the texts, (1) we tokenized the utterances with the reference Morfessor 2.0 (Virpioja et al., 2013) implementation, (2) long utterances were broken down into multiple shorter utterances with a sliding-window of 20 words and a step-size of 18 words, and (3) we split the corpus into a `train`, `dev`, and `test` split with respective proportions of 80%, 10%, and 10% of the utterances. For an overview of the final corpus statistics, including the dialog act tags, see Table 2.

For each representation type, we trained a linear *ridge-regression*[5]-based classifier with 10-fold cross-validation (on the training set) to predict the dialog act labels. Altogether, there were 42 distinct dialog acts. To simplify the probe, we only selected the 10 most frequent labels to classify, while grouping the remaining labels under a single category.

## 4   Results

The classification accuracy results are presented in Table 3. We established baseline performance as the frequency of the majority class. The only variant performing below the majority baseline was the *Random* variant ($D_{hidden} = 6$), which used means of arbitrarily assigned word vectors. The other variants achieved scores in the range of up to $15 - 21\%$ above the majority baseline. Except

---
[4]See the *Coder's manual* (Jurafsky, 1997) for more information.

[5]We opted for ridge regression, instead of logistic regression, because of high colinearity (except for LDA and LSA) across feature vectors. See correlation matrices in the appendix Section A. We used the Scikit-Learn implementation.

|          |      |         | Context |        |
|          |      | Content | 1 utt.  | 2 utt. |
| Algo.    | Dim. |         | Accuracy [%] |   |
|----------|------|---------|---------|--------|
| Maj. class |    | 35.00   |         |        |
| GLOVE    | 6    | 51.62   | 50.37   | 50.14  |
|          | 12   | 52.82   | 52.60   | 52.55  |
|          | 25   | **57.29** | 57.24 | 56.52  |
| LDA      | 6    | 51.10   | 51.16   | 51.05  |
|          | 12   | 53.32   | 52.99   | 52.73  |
|          | 25   | **56.82** | 56.36 | 56.13  |
| LSA      | 6    | 51.07   | 50.62   | 50.58  |
|          | 12   | 51.94   | 51.44   | 51.04  |
|          | 25   | 52.02   | 52.11   | **52.12** |
| Random   | 6    | 36.52   | 41.55   | 41.83  |
|          | 12   | 46.99   | 47.17   | 46.47  |
|          | 25   | **48.46** | 47.69 | 47.59  |
| Auto-enc.| 6    | 53.88   | 52.27   | 51.63  |
|          | 12   | **54.46** | 53.89 | 52.34  |
|          | 25   | 52.43   | 53.18   | 52.57  |
| Skip-Th. | 6    |         | 51.13   | 39.66  |
|          | 12   |         | 53.33   | 52.38  |
|          | 25   |         | 55.21   | **57.54** |

Table 3: Results. The three right-most columns contain the accuracy percentages for the content-only, context of distance 1, and context of distance 2 variants.

for LSA, each algorithm achieved the highest score with $D_{hidden} = 25$.

There were no big differences and no discerning patterns in the performance of the contextually projected variants.

The only case where the training regime of predicting two neighboring sentences on each side scored the best was with the end-to-end, Skip-Thought like variant, which also achieved the overall highest score.

## 5   Discussion

All tested models performed well over the majority baseline, including the averaged random word vectors which exceeded the majority baseline by a large margin when coupled with high dimensionality. One reason for this might be that, when averaging word vectors of higher dimension, there is a higher chance that some important lexical features will remain separable from the noise (also applicable to GloVe). Only in case of LSA and Auto-Encoder, the effect of dimensionality was small. For the latter, using a variational or contrastive learning set-up might allow for better scaling of

performance with capacity. Based on the patterns observed here, increasing the dimensionality increases performance on the probe, with or without context.

We did not find either of these aspects discussed in the mainstream related work (Conneau et al., 2018; Belinkov, 2022b). The question of how far increasing the dimensionality helps performance should be explored in future work.

The presented results on the dialog act tag probing task are in line with related literature in that sequence/document representations with word embeddings and classical topic modeling approaches are a tough-to-beat baseline for many neural networks (Conneau et al., 2018; Reimers and Gurevych, 2019). Our results suggest that transforming existing, content-oriented representations to context-oriented ones provides minimal to no meaningful improvement in representation. An improvement of small, but perhaps meaningful magnitude is one from the best performing Auto-Encoder to Skip-Thought variant ( 3%), suggesting that some benefit in this training scheme may be found for sentences, too.

However, our experimental design limits us in drawing conclusions about what the Skip-Thought set-up benefited from specifically: the more general sentence representations may simply be an artifact of the more difficult learning task (predicting neighboring sentences), as opposed to being a result of form disambiguation via context, the way it works with words. Future work should employ methodology that separates the contribution of the competing sources of signal more clearly.

When compared to the obvious benefits of context on learning word representations, it is striking that the benefits for learning sentence representations appear to be very small at best.

This observation is suggestive of a more serious blind-spot in NLP: because context-based representations work well for words, many researchers assume that the same holds for any level of analysis. This is akin to claiming arbitrariness of the sign at any level of language processing.

## 6 Conclusion

We presented evidence that explicitly examines the differences between content-, and context-oriented unsupervised sentence representations. While results suggest that dimensionality, rather than context, is the determining factor in building good sentence representations, end-to-end approaches based on reconstructing surrounding sentences are a promising way forward.

## Limitations

Since this work only presents evidence from one sentence task, the conclusions drawn are task-specific. Although we believe the chosen task to be fairly general, one can easily think of tasks that would likely challenge the drawn conclusions, e.g., recognition of humor or sarcasm. However, these are rather rare cases and it is questionable, where the best position of the boundary between unsupervised and task-specific lies. Additionally, the selected range of dimensions is limited to a small amount, and at larger scales the patterns of results might change.

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

# A Correlation Matrices

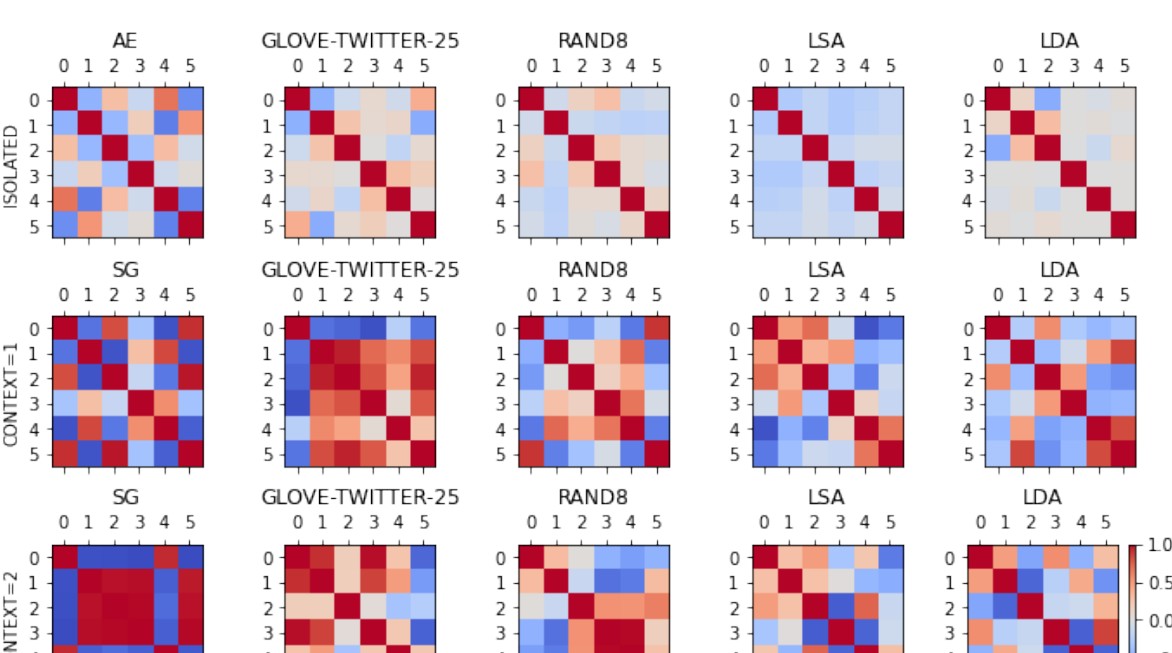

Figure 1: Correlation matrices of feature vectors across representation variants with dimension of 6.

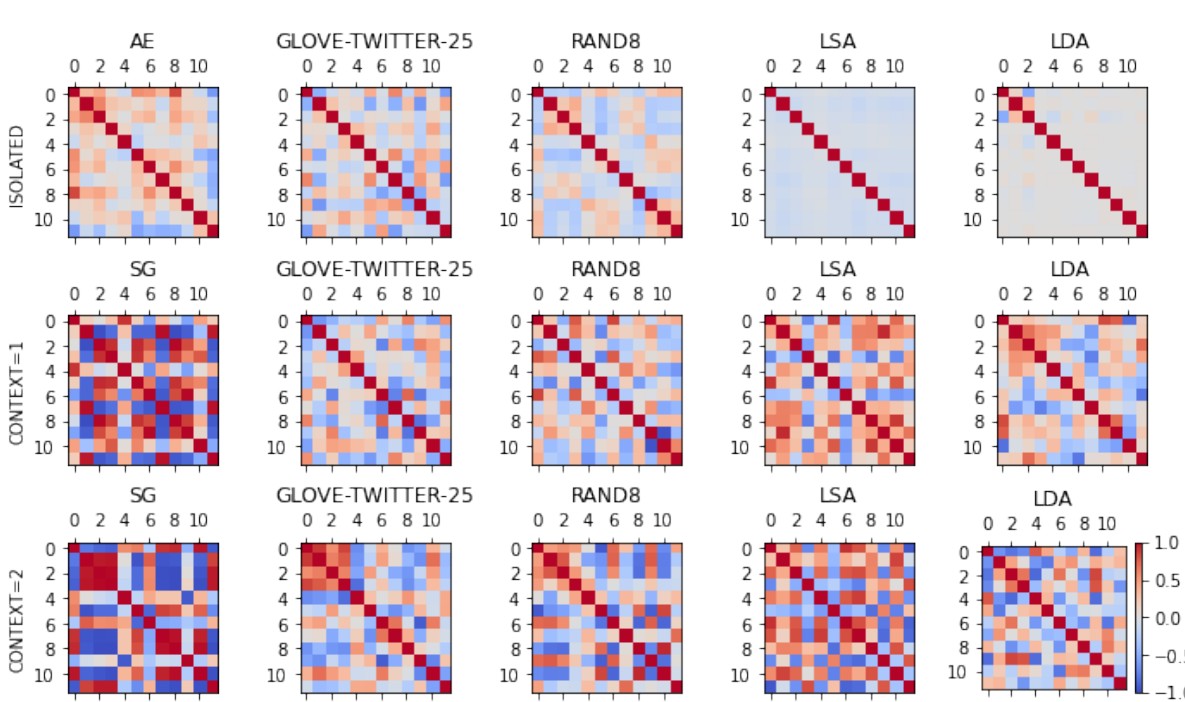

Figure 2: Correlation matrices of feature vectors across representation variants with dimension of 12.

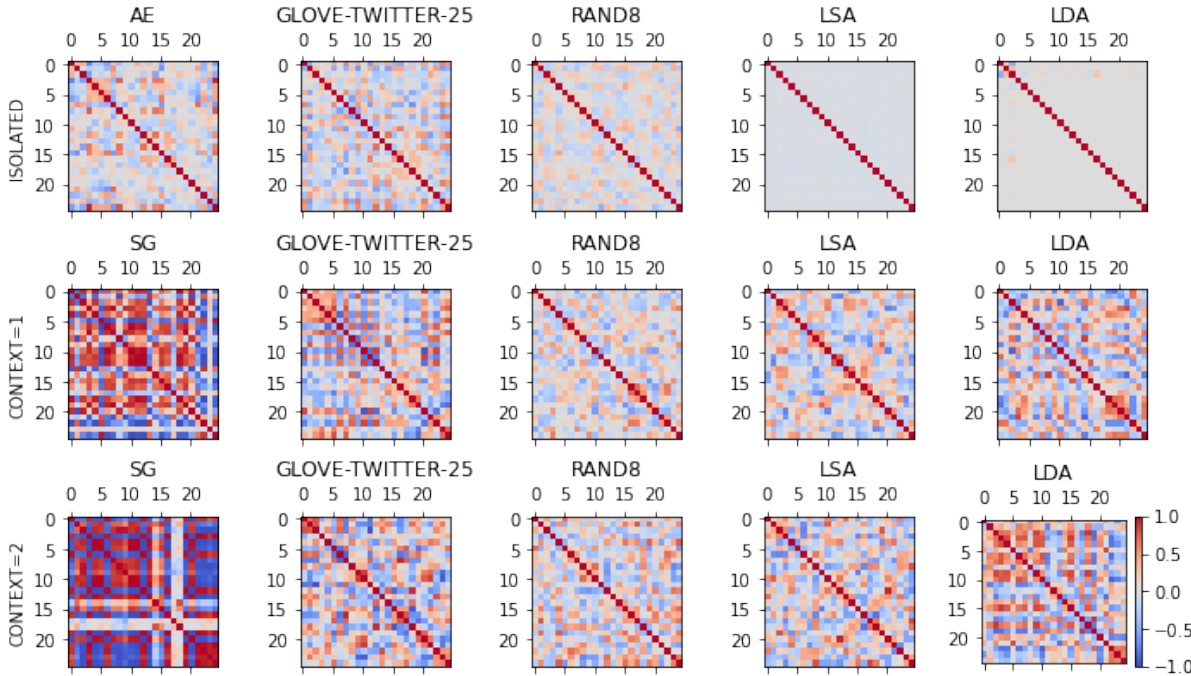

Figure 3: Correlation matrices of feature vectors across representation variants with dimension of 25.