# OpenReview forum: "Role of Context in Unsupervised Sentence Representation Learning: the Case of Dialog Act Modeling"
_EMNLP/2023/Conference — EMNLP 2023 Findings_

### Official Review · Reviewer_TQzd · 2023-08-03

**Soundness:** 4

**Excitement:**

4: Strong: This paper deepens the understanding of some phenomenon or lowers the barriers to an existing research direction.

**Missing References:**

Reading Table 3, with context = 0 meaning content-based representation, is not very intuitive.


**Paper Topic And Main Contributions:**

The paper assess the gains of using sentence representations that are learned to be predictive of the discourse context, focusing on the case of dialog act classification. They train linear probing classifier from sentence representations obtained based on sentence content only or also based on context. The results point to only minimal improvements of context-based methods, challenging the assumption that such methods are beneficial for sentence representations only because beneficial for word representations.

**Questions For The Authors:**

1. Do you have any insights or intuitions as to why 1) dimensionality would matter so much, and 2) context-based methods do not benefit sentences as much as words?

2. Do qualitative inspections of predicted dialog act assignment show patterns of what context-based methods capture well and content-based do not, and viceversa?

**Reasons To Accept:**

The paper provides a focused contribution that is well-justified and discussed. The points made are original and well explained.

The experiments provide initial results challenging a common methodological assumption in the field, that can give rise to further explorations in this area.

**Reasons To Reject:**

The scope of the empirical results is limited to a single task and corpus.

Missing insights from qualitative analyses of the predictions.

**Reproducibility:**

4: Could mostly reproduce the results, but there may be some variation because of sample variance or minor variations in their interpretation of the protocol or method.

**Reviewer Confidence:**

4: Quite sure. I tried to check the important points carefully. It's unlikely, though conceivable, that I missed something that should affect my ratings.

---

> ### Author Rebuttal · Authors · 2023-08-29
>
> Dear reviewer, thank you for your feedback.
>
> Regarding the importance of dimensionality (same answer as the one to the reviewer wEG6), it seems to be very case-specific. For the approaches that aggregate word-vectors, it might be that, as the “word signature” gets bigger, the chance increases that a presence of a certain word (i.e. lexical feature) can be more easily recovered from the final representation. In case of the auto-encoder, the seeming failure to scale with dimension may be due to its naive implementation (variational loss or a de-noising setup would probably learn more robustly). The Skip-Thought setup, an inherently more difficult task compared to the AE, benefited from higher dimension most when context was 2 (the most difficult version of the task). An alternative explanation to why this approach provided better probe scores is simply that neural network-based unsupervised representation learning scales best when (1) the representation capacity increases while (2) the task difficulty increases. To give more conclusive answers about the role of context, a further exploration would be needed, possibly including experiments with masking/denoising/contrastive learning, etc.
>
> Regarding why context-based methods do not benefit sentences as much as words, we conjecture that this is related to structural characteristics of language. In the introduction, we pointed to Saussure's arbitrariness of the sign: word form by itself does not encode meaning. In other words, semantic similarity between two words cannot be predicted based on the individual components of those words (sounds or characters). For sentences, this arbitrariness does not seem to hold to the same extent: words appearing in a sentence do contribute to the sentence's meaning and having words in common will contribute to sentence similarity in a way that having letters in common will not for words.
>
> Regarding qualitative inspections of dialog act assignment: When we looked at the confusion matrices, we did notice some differences.
> Comparison between the models trained with auto-encoding (AE), vs. skip-thought (ST; =reconstructing neighbouring utterances), specifically AE dim=12 and no context, vs. ST dim=25 and context 1 (two models that achieved similar performance: 54.46 vs. 55.21). First, AE correctly identified more “Acknowledge (Backchannel); B” utterances, which are typically very short (e.g. “uh-huh”). On one hand, such an utterance is probably not very informative in terms of what could precede it or come next, so the contextual representation probably encodes the same information as other utterances that are de facto noise. On the other hand, for the AE, encoding short content like “uh-huh” is likely an easy task. Second, the ST correctly distinguished between more SD (statement-non-opinion) vs. SV (statement-opinion) labeled utterances (with the AE, these classifications collapsed to the majority class SD). Upon further inspection, we didn’t notice a marked contextual signal that would explain this difference, so this might in fact be a simple artefact of a better/harder learning objective in the case of ST. Lastly, when comparing the context-adjusted representations to their original (content-based) counter-parts (e.g. LDA, GloVe, LSA), we did not see any qualitative differences.
>
> These observations are indicative of a methodological challenge that may be worth addressing in future work: when training neural network-based models with a reconstruction objective and comparing identical architectures optimised with differently sampled target signal, the differences in the representations might still be confounded between the usefulness of the target signal and the degree to which the training regimen can actually extract the useful information.

---

### Official Review · Reviewer_F3K4 · 2023-08-09

**Soundness:** 3

**Excitement:**

3: Ambivalent: It has merits (e.g., it reports state-of-the-art results, the idea is nice), but there are key weaknesses (e.g., it describes incremental work), and it can significantly benefit from another round of revision. However, I won't object to accepting it if my co-reviewers champion it.

**Missing References:**

- SimCSE: Simple Contrastive Learning of Sentence Embeddings https://arxiv.org/abs/2104.08821
- DiffCSE: Difference-based Contrastive Learning for Sentence Embeddings https://arxiv.org/abs/2204.10298
- Learning dialogue representations from consecutive utterances https://arxiv.org/abs/2205.13568
- TOD-BERT: Pre-trained Natural Language Understanding for Task-Oriented Dialogue https://arxiv.org/abs/2004.06871
- TaDSE: Template-aware Dialogue Sentence Embeddings https://arxiv.org/abs/2305.14299
- FutureTOD: Teaching Future Knowledge to Pre-trained Language Model for Task-Oriented Dialogue https://arxiv.org/abs/2306.10315

**Paper Topic And Main Contributions:**

The paper experiments with dialog act probing task to compare content and context-oriented sentence representations on the SwDA dataset. The results suggest no clear benefit between context-based sentence representations over content-based sentence representations. Increasing the dimensionality of the sentence vectors helps in all approaches.

**Reasons To Accept:**

1. The paper is well written.
2. Experimental method is sound.

**Reasons To Reject:**

1. It's a short paper, but it may benefit from an evaluation on an additional task (i.e. NLI) which is more common.
2. Mentioning the relation between dimensionality and performance as a contribution is too marginal. In addition, the discussion in regard to LSA and Auto-Encoder seems insufficient.
3. The paper does not cite or perform experiments with LM-based unsupervised sentence representation learning methods, starting with SimCSE and dialogue embeddings such as DSE which are state-of-the-art in unsupervised sentence embeddings. They create embeddings with BERT models. The related works section should be revised (citations below), with more experiments.
4. Appendix A should provide an analysis of the correlation matrix, instead of a blank space.

**Reproducibility:**

3: Could reproduce the results with some difficulty. The settings of parameters are underspecified or subjectively determined; the training/evaluation data are not widely available.

**Reviewer Confidence:**

4: Quite sure. I tried to check the important points carefully. It's unlikely, though conceivable, that I missed something that should affect my ratings.

---

> ### Author Rebuttal · Authors · 2023-08-29
>
> Dear reviewer, thank you for your feedback. For this study, we deliberately intended to use a task (and dataset) from a narrow domain with only few semantic categories, such that the experiment can be performed with little compute from scratch. Indeed, we did not consider some popular NLP tasks for this purpose, e.g., NLI, SICK, SST(2)), because such representations would have to be way more “universal” to be useful (in practice, most likely implemented via a (large) language model, requiring more expensive experiments). We recognise that there are more expressive, state-of-the-art models that achieve high scores on these tasks, but we did not regard the model’s overall achieved performance as compromising our objective of contrasting the two sources of signal. Nonetheless, we acknowledge that we should mention them in the related work section.

---

### Official Review · Reviewer_wEG6 · 2023-08-10

**Soundness:** 4

**Excitement:**

4: Strong: This paper deepens the understanding of some phenomenon or lowers the barriers to an existing research direction.

**Paper Topic And Main Contributions:**

The authors investigated to what extent context plays a role in using unsupervised methods for learning sentence representation.
They used a corpus made up of utterances of telephone conversations with dialog act labels. First, they trained a bidirectional LSTM to extract content-based representations (content vectors) of sentences. Then, they trained another Skip-Thought-like model for creating for each utterance the context-based representations of its surrounding sentences.
Finally, they evaluate such representations with a probing task by testing the models in predicting the correct dialog act labels.

**Questions For The Authors:**

1. What are your thoughts about the issue of dimensionality? Why do you think it can influence the performance so much?

**Reasons To Accept:**

The paper represents a valuable attempt to identify to what extent context plays a role in using unsupervised methods for learning sentence representation. At the same time, this study is a good way to attest how these models adopt dynamics that move away from human-like procedures, being influenced by other parameters. From this point of view, the findings about the importance of dimensionality and how it can improve performance, with or without context, is very interesting and it can be a good starting point for further investigations. The main objective of the paper is very clear and so are the final results obtained.

**Reasons To Reject:**

Probably the greatest weakness of this research may lie in the corpus used. The size and heterogeneity of the data are adequate for the purpose of this study, but it seems that an analysis of each conversation (and its constituent utterances) was not carried out in order to understand whether the contexts (represented by the surrounding sentences) could really be informative. This could therefore influence the models in favouring a content-based rather than context-based representation of sentences.
For this reason, it would be preferable to carry out an a priori study on the dataset to make it more cognitively grounded. For example, human judgements could be collected to make sure that contexts are really informative in identifying the utterance under investigation.

**Reproducibility:**

4: Could mostly reproduce the results, but there may be some variation because of sample variance or minor variations in their interpretation of the protocol or method.

**Reviewer Confidence:**

3: Pretty sure, but there's a chance I missed something. Although I have a good feel for this area in general, I did not carefully check the paper's details, e.g., the math, experimental design, or novelty.

**Typos Grammar Style And Presentation Improvements:**

The concept of 'teacher forcing' at line 209 is taken for granted, but I think you should explain it, maybe with a footnote. Moreover, I think that there is a typo at line 301.

---

> ### Author Rebuttal · Authors · 2023-08-29
>
> Dear reviewer, thank you for your feedback. We assumed that contextual information can be informative for this task mainly based on various other studies that develop classifiers (at least partially) based on it, for example [1, 2, 3]. We also checked the reduction in entropy when “guessing” a single isolated label vs. predicting it given the previous label (a “bigram” setting), and the results were $H(currentlabel) = 2.1bit$ and $H(currentlabel | previouslabel) = 1.86bit$. Such a context-enhanced baseline would, at the least, result in higher chance-level accuracy.
>
> Regarding the dimensionality, it seems to be very case-specific. For the approaches that aggregate word-vectors, it might be that, as the “word signature” gets bigger, the chance increases that a presence of a certain word (i.e. lexical feature) can be more easily recovered from the final representation. In case of the auto-encoder, the seeming failure to scale with dimension may be due to its naive implementation (variational loss or a de-noising setup would probably learn more robustly). The Skip-Thought setup, an inherently more difficult task compared to the AE, benefited from higher dimension most when context was 2 (the most difficult version of the task). An alternative explanation to why this approach provided better probe scores is simply that neural network-based unsupervised representation learning scales best when (1) the representation capacity increases while (2) the task difficulty increases. To give more conclusive answers about the role of context, a further exploration would be needed, possibly including experiments with masking/denoising/contrastive learning, etc.
>
> [1] Dialogue Act Classification with Context-Aware Self-Attention (Raheja, Tetreault, 2019)
>
> [2] Recurrent convolutional neural networks for discourse compositionality (Kalchbrenner, Brunson, 2013)
>
> [3] Multi-level gated recurrent neural network for dialog act classification (Li, Wu, 2016)

---

### Meta-Review · Area_Chair_wvJs · 2023-09-15

**Recommendation:** 4

**Metareview:**

This short paper investigates the contribution of context in unsupervised methods for building sentence representations, by comparing content-based methods (e.g. topic modeling, word vector averaging, LSTM autoencoders) with their context-based version, which was obtained by using the same vectors as the input for a Skip Thought-like training regime. The evaluation, conducted on a dialog act tagging task on the Switchboard corpus via a linear probing classifier, reveal that the advantage of the latter training regime is not significant, while there is a clear benefit in the simple increase of the dimensionality of the sentence vector.

All the reviewers agree that this is a clear and well-conducted study. On the weak side, I tend to agree with Reviewer 2, who pointed out that the contribution of this paper is incremental, and it would be beneficial to add some extra experiments with more recent sentence representations method.

---

### Decision · Program_Chairs · 2023-10-07

**Decision:**

Accept-Findings

**Comment:**

This short paper investigates the contribution of context in unsupervised methods for building sentence representations, by comparing content-based methods (e.g. topic modeling, word vector averaging, LSTM autoencoders) with their context-based version, which was obtained by using the same vectors as the input for a Skip Thought-like training regime. The evaluation, conducted on a dialog act tagging task on the Switchboard corpus via a linear probing classifier, reveal that the advantage of the latter training regime is not significant, while there is a clear benefit in the simple increase of the dimensionality of the sentence vector.

All the reviewers agree that this is a clear and well-conducted study. On the weak side, I tend to agree with Reviewer 2, who pointed out that the contribution of this paper is incremental, and it would be beneficial to add some extra experiments with more recent sentence representations method.